# Training Generative Adversarial Networks from Incomplete Observations using Factorised Discriminators

**Daniel Stoller**
Queen Mary University
London, UK
d.stoller@qmul.ac.uk

**Sebastian Ewert**
Spotify
Berlin, Germany
sewert@spotify.com

**Simon Dixon**
Queen Mary University
London, UK
s.e.dixon@qmul.ac.uk

## Abstract

Generative adversarial networks (GANs) have shown great success in applications such as image generation and inpainting. However, they typically require large datasets, which are often not available, especially in the context of prediction tasks such as image segmentation that require labels. Therefore, methods such as the CycleGAN use more easily available unlabelled data, but do not offer a way to leverage additional labelled data for improved performance. To address this shortcoming, we show how to factorise the joint data distribution into a set of lower-dimensional distributions along with their dependencies. This allows splitting the discriminator in a GAN into multiple "sub-discriminators" that can be independently trained from incomplete observations. Their outputs can be combined to estimate the density ratio between the joint real and the generator distribution, which enables training generators as in the original GAN framework. We apply our method to image generation, image segmentation and audio source separation, and obtain improved performance over a standard GAN when additional incomplete training examples are available. For the Cityscapes segmentation task in particular, our method also improves accuracy by an absolute $14.9\%$ over CycleGAN while using only 25 additional paired examples.

## 1 Introduction

In generative adversarial networks (GANs) (Goodfellow et al., 2014) a generator network is trained to produce samples from a given target distribution. To achieve this, a discriminator network is employed to distinguish between "real" samples from the dataset and "fake" samples from the generator network. The discriminator's feedback is used by the generator to improve its output. While GANs have become highly effective at synthesising realistic examples even for complex data such as natural images (Radford et al., 2015; Karras et al., 2018), they typically rely on large training datasets. These are not available in many cases, especially for prediction tasks such as audio source separation (Stoller et al., 2018) or image-to-image translation (Zhu et al., 2017). Instead, one often encounters many incomplete observations, such as unpaired images in image-to-image translation, or isolated source recordings in source separation. However, standard GANs cannot be trained with these observations. Recent approaches that work with unpaired data can not make use of additional paired data (Zhu et al., 2017) or lead to computational overhead due to additional generators and discriminators that model the inverse of the mapping of interest (Almahairi et al., 2018; Gan et al., 2017). For training the generator, multiple losses are combined whose interactions are not clear and that do not guarantee that the generator converges to the desired distribution.

In this paper, we adapt the standard GAN framework to enable training predictive models with both paired and unpaired data as well as generative models with incomplete observations. To achieve this, we split the discriminator into multiple "marginal" discriminators, each modelling a separate set of dimensions of the input. As this modification on its own would ignore any dependencies between these parts, we incorporate two additional "dependency discriminators", each focusing only on inter-part relationships. We show how the outputs from these marginal and dependency discriminators can be recombined and used to estimate the same density ratios as in the original

GAN framework – which enables training any generator network in an unmodified form. In contrast to previous GANs, our approach only requires full observations to train the smaller dependency discriminator and can leverage much bigger, simpler datasets to train the marginal discriminators, which enables the generator to model the marginal distributions more accurately. Additionally, prior knowledge about the marginals and dependencies can be incorporated into the architecture of each discriminator. Deriving from first principles, we obtain a consistent adversarial learning framework without the need for extra losses that rely on more assumptions or conflict with the GAN objective.

In our experiments, we apply our approach ("FactorGAN") [1] to two image generation tasks (Sections 4.1 and 4.2), image segmentation (Section 4.3) and audio source separation (Section 4.4), and observe improved performance in missing data scenarios compared to a GAN. For image segmentation, we also compare to the CycleGAN (Zhu et al., 2017), which does not require images to be paired with their segmentation maps. By leveraging both paired and unpaired examples with a unified adversarial objective, we achieve a substantially higher segmentation accuracy even with only 25 paired samples than GAN and CycleGAN models.

## 2 METHOD

After a brief summary of GANs in Section 2.1, we introduce our method from a missing data perspective in Section 2.2, before extending it to conditional generation (Section 2.3) and the case of independent outputs (Section 2.4).

### 2.1 GENERATIVE ADVERSARIAL NETWORKS

To model a probability distribution $p_x$ over $\mathbf{x} \in \mathbb{R}^d$, we follow the standard GAN framework and introduce a generator model $G_\phi : \mathbb{R}^n \to \mathbb{R}^d$ that maps an $n$-dimensional input $\mathbf{z} \sim p_z$ to a $d$-dimensional sample $G_\phi(\mathbf{z})$, resulting in the generator distribution $q_x$. To train $G_\phi$ such that $q_x$ approximates the real data density $p_x$, a discriminator $D_\theta : \mathbb{R}^d \to (0, 1)$ is trained to estimate whether a given sample is real or generated:

$$\arg \max_\theta \ \mathbb{E}_{\mathbf{x} \sim p_x} \log D_\theta(\mathbf{x}) + \mathbb{E}_{\mathbf{x} \sim q_x} \log(1 - D_\theta(\mathbf{x})). \tag{1}$$

In the non-parametric limit (Goodfellow et al., 2014), $D_\theta(\mathbf{x})$ approaches $\tilde{D}(\mathbf{x}) := \frac{p_x(\mathbf{x})}{p_x(\mathbf{x}) + q_x(\mathbf{x})}$ at every point $\mathbf{x}$. The generator is updated based on the discriminator's estimate of $\tilde{D}(\mathbf{x})$. In this paper, we use the alternative loss function for $G_\phi$ as proposed by Goodfellow et al. (2014):

$$\arg \max_\theta \ \mathbb{E}_{\mathbf{z} \sim p_z} \log D_\theta(G_\phi(\mathbf{z})). \tag{2}$$

### 2.2 ADAPTATION TO MISSING DATA

In the following we consider the case that incomplete observations are available in addition to our regular dataset (i.e. simpler yet larger datasets). In particular, we partition the set of $d$ input dimensions of $\mathbf{x}$ into $K$ ($2 \leq K \leq d$) non-overlapping subsets $\mathcal{D}_1, \ldots, \mathcal{D}_K$. For each $i \in \{1, \ldots, K\}$, an incomplete ("marginal") observation $\mathbf{x}^i$ can be drawn from $p_x^i$, which is obtained from $p_x$ after marginalising out all dimensions not in $\mathcal{D}_i$. Analogously, $q_x^i$ denotes the $i$-th marginal distribution of the generator $G_\phi$. Next, we extend the existing GAN framework such we can employ the additional incomplete observations. In this context, a main hurdle is that a standard GAN discriminator is trained with samples from the full joint $p_x$. To eliminate this restriction, we note that $\tilde{D}(\mathbf{x})$ can be mapped to a "joint density ratio" $\frac{p_x(\mathbf{x})}{q_x(\mathbf{x})}$ by applying the bijective function $h : [0, 1) \to \mathbb{R}^+, h(a) = -\frac{a}{a-1}$. For our approach, we exploit that this joint density ratio can be factorised into a product of density ratios:

$$h(\tilde{D}(\mathbf{x})) = \frac{p_x(\mathbf{x})}{q_x(\mathbf{x})} = \frac{c_P(\mathbf{x})}{c_Q(\mathbf{x})} \prod_{i=1}^{K} \frac{p_x^i(\mathbf{x}^i)}{q_x^i(\mathbf{x}^i)} \text{ with}$$

$$c_P(\mathbf{x}) = \frac{p_x(\mathbf{x})}{\prod_{i=1}^{K} p_x^i(\mathbf{x}^i)} \text{ and } c_Q(\mathbf{x}) = \frac{q_x(\mathbf{x})}{\prod_{i=1}^{K} q_x^i(\mathbf{x}^i)}. \tag{3}$$

---

[1] Code available at https://github.com/f90/FactorGAN

Each "marginal density ratio" $\frac{p_x^i(\mathbf{x}^i)}{q_x^i(\mathbf{x}^i)}$ captures the generator's output quality for one marginal variable $\mathbf{x}^i$, while the $c_P$ and $c_Q$ terms describe the dependency structure between marginal variables in the real and generated distribution, respectively. Note that our theoretical considerations assume that the densities $p_x$ and $q_x$ are non-zero everywhere. While this might not be fulfilled in practice, our implementation does not directly compute density ratios and instead relies on the same assumptions as Goodfellow et al. (2014). We can estimate each density ratio independently by training a "sub-discriminator" network, and combine their outputs to estimate $\tilde{D}(\mathbf{x})$, as shown below.

**Estimating the marginal density ratios:**   To estimate $\frac{p_x^i(\mathbf{x}^i)}{q_x^i(\mathbf{x}^i)}$ for each $i \in \{1, \ldots, K\}$, we train a "marginal discriminator network" $D_{\theta_i} : \mathbb{R}^{|\mathcal{D}_i|} \to (0, 1)$ with parameters $\theta_i$ to determine whether a marginal sample $\mathbf{x}^i$ is real or generated following the GAN discriminator loss in Equation (1) [2]. This allows making use of the additional incomplete observations. In the non-parametric limit, $D_{\theta_i}(\mathbf{x}^i)$ will approach $\tilde{D}_i(\mathbf{x}^i) := \frac{p_x^i(\mathbf{x}^i)}{p_x^i(\mathbf{x}^i)+q_x^i(\mathbf{x}^i)}$, so that we can use $h(D_{\theta_i}(\mathbf{x}^i))$ as an estimate of $\frac{p_x^i(\mathbf{x}^i)}{q_x^i(\mathbf{x}^i)}$.

**Estimation of $c_P(\mathbf{x})$ and $c_Q(\mathbf{x})$:**   Note that $c_P$ and $c_Q$ are also density ratios, this time containing a distribution over $\mathbf{x}$ in both the numerator and denominator – the main difference being that in the latter the individual parts $\mathbf{x}^i$ are independent from each other. To approximate the ratio $c_P$, we can apply the same principles as above and train a "p-dependency discriminator" $D_{\theta_P}^P : \mathbb{R}^d \to (0, 1)$ to distinguish samples from the two distributions, i.e. to discriminate real joint samples from samples where the individual parts are real but were drawn independently of each other (i.e. the individual parts might not originate from the same real joint sample). Again, in the non-parametric limit, its response approaches $\tilde{D}^P(\mathbf{x}) := \frac{p_x(\mathbf{x})}{p_x(\mathbf{x})+\prod_{i=1}^K p_x^i(\mathbf{x}^i)}$ and thus $c_P$ can be approximated via $h \circ D_{\theta_P}^P$. Analogously, the $c_Q$ term is estimated with a "q-dependency discriminator" $D_{\theta_Q}^Q$ – here, we compare joint generator samples with samples where the individual parts were shuffled across several generated samples (to implement the independence assumption).

**Joint discriminator sample complexity:**   In contrast to $c_Q$, where the generator provides an infinite number of samples, estimating $c_P$ without overfitting to the limited number of joint training samples can be challenging. While standard GANs suffer from the same difficulty, our factorisation into specialised sub-units allows for additional opportunities to improve the sample complexity. In particular, we can design the architecture of the p-dependency discriminator to incorporate prior knowledge about the dependency structure[3].

**Combining the discriminators:**   As the marginal and the p- and q-dependency sub-discriminators provide estimates of their respective density ratios, we can multiply them and apply $h^{-1}$ to obtain the desired ratio $\tilde{D}(\mathbf{x})$, following Equation (3). This can be implemented in a simple and stable fashion using a linear combination of pre-activation sub-discriminator outputs followed by a sigmoid (see Section A.4 for details and proof). The time for a generator update step grows linearly with the number of marginals $K$, assuming the time to update each of the $K$ marginal discriminators remains constant.

## 2.3   ADAPTATION TO CONDITIONAL GENERATION

Conditional generation, such as image segmentation or inpainting, can be performed with GANs by using a generator $G_\phi$ that maps a conditional input $\mathbf{x}^1$ and noise to an output $\mathbf{x}^2$, resulting in an output probability $q_\phi(\mathbf{x}^2|\mathbf{x}^1)$.

When viewing $\mathbf{x}^1$ and $\mathbf{x}^2$ as parts of a joint variable $\mathbf{x} := (\mathbf{x}^1, \mathbf{x}^2)$ with distribution $p_x$, we can also frame the above task as matching $p_x$ to the joint generator distribution $q_x(\mathbf{x}) := p_x^1(\mathbf{x}^1)q_\phi(\mathbf{x}^2|\mathbf{x}^1)$. In a standard conditional GAN, the discriminator is asked to distinguish between joint samples from $p_x$ and $q_x$, which requires *paired* samples from $p_x$ and is inefficient as the inputs $\mathbf{x}^1$ are the same in

---

[2]Samples are drawn from $p_x^i$ and $q_x^i$ instead of $p_x$ and $q_x$, respectively.

[3]If only certain features of a marginal variable influence the dependencies, we can limit the input to the p-dependency discriminator to these features instead of the full marginal sample to prevent overfitting.

both $p_x$ and $q_x$. In contrast, applying our factorisation principle from Equation (3) to $\mathbf{x}^1$ and $\mathbf{x}^2$ (for the special case $K = 2$) yields

$$\frac{p_x(\mathbf{x})}{q_x(\mathbf{x})} = \frac{\frac{p_x(\mathbf{x})}{p_x^1(\mathbf{x}^1)p_x^2(\mathbf{x}^2)}}{\frac{q_x(\mathbf{x})}{q_x^1(\mathbf{x}^1)q_x^2(\mathbf{x}^2)}} \frac{p_x^2(\mathbf{x}^2)}{q_x^2(\mathbf{x}^2)} = \frac{c_P(\mathbf{x})}{c_Q(\mathbf{x})} \frac{p_x^2(\mathbf{x}^2)}{q_x^2(\mathbf{x}^2)}, \tag{4}$$

suggesting the use of a p- and a q-dependency discriminator to model the input-output relationship, and a marginal discriminator over $\mathbf{x}^2$ that matches aggregate generator predictions from $q_x^2$ to real output examples from $p_x^2$. Note that we do not need a marginal discriminator for $\mathbf{x}^1$, which increases computational efficiency. This adaptation can also involve additionally partitioning $\mathbf{x}^2$ into multiple partial observations as shown in Equation 3.

### 2.4 ADAPTION TO INDEPENDENT MARGINALS

In case the marginals can be assumed to be completely independent, one can remove the p-dependency discriminator from our framework, since $c_P(\mathbf{x}) = 1$ for all inputs $\mathbf{x}$. This approach can be useful in the conditional setting, when each output is related to the input but their marginals are independent from each other. In this setting, our method is related to adversarial ICA (Brakel & Bengio, 2017). Note that the q-dependency discriminator still needs to be trained on the full generator outputs if the generator should not introduce unwanted dependencies between the marginals.

### 2.5 FURTHER EXTENSIONS

There are many more ways of partitioning the joint distribution into marginals. We discuss two additional variants (*Hierarchical and auto-regressive FactorGANs*) of our approach in Section A.3.

## 3 RELATED WORK

For conditional generation, "CycleGAN" (Zhu et al., 2017) exploits unpaired samples by assuming a one-to-one mapping between the domains and using bidirectional generators (along with Gan et al. (2017)), while FactorGAN makes no such assumptions and instead uses paired examples to learn the dependency structure. Almahairi et al. (2018) and Tripathy et al. (2018) learn from paired examples with an additional reconstruction-based loss, but use a sum of many different loss terms which have to be balanced by additional hyper-parameters. Additionally, it can not be applied to generation tasks with missing data or prediction tasks with multiple outputs. Brakel & Bengio (2017) perform independent component analysis in an adversarial fashion using a discriminator to identify correlations. Similarly to our q-dependency discriminator, the separator outputs are enforced to be independent, but our method is fully adversarial and can model arbitrary dependencies with the p-dependency discriminator. GANs were also used for source separation, but dependencies were either ignored (Zhang et al., 2017) or modelled with an additional L2 loss (Stoller et al., 2018) that supports only deterministic separators.

Pu et al. (2018) use GANs for joint distribution modelling by training a generator for each possible factorisation of the joint distribution, but this requires $K!$ generators for $K$ marginals, whereas we assume either all parts or exactly one part of the variable of interest is observed to avoid functional redundancies between the different networks. Karaletsos (2016) propose adversarial inference on local factors of a high-dimensional joint distribution and factorise both generator and discriminator based on independence assumptions given by a Bayesian network, whereas we keep a joint sample generator and model all dependencies. Finally, Yoon et al. (2018) randomly mask the inputs to a GAN generator so it learns to impute missing values, whereas our generator aims to learn a transformation where inputs are fully observed.

## 4 EXPERIMENTS

To validate our method, we compare our FactorGAN with the regular GAN approach, both for unsupervised generation as well as supervised prediction tasks. For the latter, we also compare to the CycleGAN (Zhu et al., 2017) as an unsupervised baseline. To investigate whether FactorGAN makes

efficient use of all observations, we vary the proportion of the training samples available for joint sampling (paired), while using the rest to sample from the marginals (unpaired). We train all models using a single NVIDIA GTX 1080 GPU.

**Training procedure**  For stable training, we employ spectral normalisation (Miyato et al., 2018) on each discriminator network to ensure they satisfy a Lipschitz condition. Since the overall output used for training the generator is simply a linear combination of the individual discriminators (see Section A.4), the generator gradients are also constrained in magnitude accordingly. Unless otherwise noted, we use an Adam optimiser with learning rate $10^{-4}$ and a batch size of 25 for training all models. We perform two discriminator updates after each generator update.

## 4.1  PAIRED MNIST

Our first experiment will involve "Paired MNIST", a synthetic dataset of low complexity whose dependencies between marginals can be easily controlled. More precisely, we generate a paired version of the original MNIST dataset[4] by creating samples that contain a pair of vertically stacked digit images. With a probability of $\lambda$, the lower digit chosen during random generation is the same as the upper one, and different otherwise. For FactorGAN, we model the distributions of upper and lower digits as individual marginal distributions ($K = 2$).

**Experimental setup**  We compare the normal GAN with our FactorGAN, also including a variant without p-dependency discriminator that assumes marginals to be independent ("FactorGAN-no-cp"). We conduct the experiment with $\lambda = 0.1$ and $\lambda = 0.9$ and also vary the amount of training samples available in paired form, while keeping the others as marginal samples only usable by FactorGAN. For both generators and discriminators, we used simple multi-layer perceptrons (Tables 1 and 2).

To evaluate the quality of generated digits, we adopt the "Frecht Inception Distance" (FID) as metric (Heusel et al., 2017). It is based on estimating the distance between the distributions of hidden layer activations of a pre-trained Imagenet object detection model for real and fake examples. To adapt the metric to MNIST data, we pre-train a classifier to predict MNIST digits (see Table 3) on the training set for 20 epochs, obtaining a test accuracy of $98\%$. We input the top and bottom digits in each sample separately to the classifier and collect the activations from the last hidden layer (FC1) to compute FIDs for the top and bottom digits, respectively. We use the average of both FIDs to measure the overall output quality of the marginals (lower value is better).

Since the only dependencies in the data are digit correlations controlled by $\lambda$, we can evaluate how well FactorGAN models these dependencies. We compute $p_D(D_t, D_b)$ as the probability for a real sample to have digit $D_t \in \{0, \ldots, 9\}$ at the top and digit $D_b \in \{0, \ldots, 9\}$ at the bottom, along with marginal probabilities $p_D^t(D_t)$ and $p_D^b(D_b)$ (and analogously $q_D(D_t, D_b)$ for generated data). Since we do not have ground truth digit labels for the generated samples, we instead use the class predicted by the pre-trained classifier. We encode the dependency as a ratio between a joint and the product of its marginals, where the ratios for real and generated data are ideally the same. Therefore, we take their absolute difference for all digit combinations as evaluation metric (lower is better):

$$d_{\text{dep}} = \frac{1}{100} \sum_{D_t=0}^{9} \sum_{D_b=0}^{9} \left| \frac{p_D(D_t, D_b)}{p_D^t(D_t) p_D^b(D_b)} - \frac{q_D(D_t, D_b)}{q_D^t(D_t) q_D^b(D_b)} \right|. \tag{5}$$

Note that the metric computes how well dependencies in the real data are modelled by a generator, but not whether it introduces any additional unwanted dependencies such as top and bottom digits sharing stroke thickness, and thus presents only a necessary condition for a good generator.

**Results**  The results of our experiment are shown in Figure 1. Since FactorGAN-no-cp trains on all samples independently of the number of paired observations, both FID and $d_{\text{dep}}$ are constant. As expected, FactorGAN-no-cp delivers good digit quality, and performs well for $\lambda = 0.1$ (as it assumes independence) and badly for $\lambda = 0.9$ with regards to dependency modelling.

FactorGAN outperforms GAN with small numbers of paired samples in terms of FID by exploiting the additional unpaired samples, although this gap closes as both models eventually have access to

---

[4]http://yann.lecun.com/exdb/mnist/

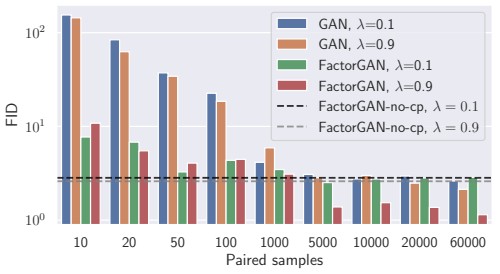 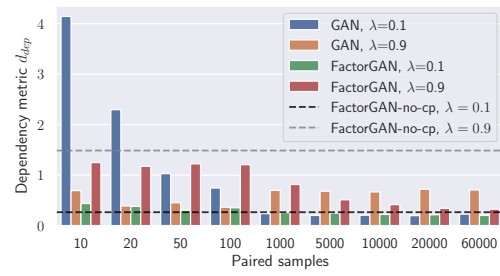

(a) FID value, averaged over both digits         (b) Dependency metric

Figure 1: Performance with different numbers of paired training samples and settings for $\lambda$ compared between GAN and FactorGAN with and without dependency modelling.

the same amount of data. FactorGAN also consistently improves in modelling the digit dependencies with an increasing number of paired observations. For $\lambda = 0.1$, this also applies to the normal GAN, although its performance is much worse for small sample sizes as it introduces unwanted digit dependencies. Additionally, its performance appears unstable for $\lambda = 0.9$, where it achieves the best results for a small number of paired examples. Further improvements in this setting could be gained by incorporating prior knowledge about the nature of these dependencies into the p-dependency discriminator to increase its sample efficiency, but this is left for future work.

## 4.2 IMAGE PAIR GENERATION

In this section, we use GAN and FactorGAN for generating pairs of images in an unsupervised way to evaluate how well FactorGAN models more complex data distributions.

**Datasets** We use the "Cityscapes" dataset (Cordts et al., 2016) and the "Edges2Shoes" dataset (Isola et al., 2016). To keep the outputs in a continuous domain, we treat the segmentation maps in the Cityscapes dataset as RGB images, instead of a set of discrete categorical labels. Each input and output image is downsampled to $64 \times 64$ pixels as a preprocessing step to reduce computational complexity and to ensure stable GAN training.

**Experimental setup** We define the distributions of input as well as output images as marginal distributions. Therefore, FactorGAN uses two marginal discriminators and a p- and q-dependency discriminator. All discriminators employ a convolutional architecture shown in Table 5 with $W = 6$ and $H = 6$. To control for the impact of discriminator size, we also train a GAN with twice the number of filters in each discriminator layer to match its size with the combined size of the FactorGAN discriminators. The same convolutional generator shown in Table 4 is used for GAN and FactorGAN. Each image pair is concatenated along the channel dimension to form one sample, so that $C = 6$ for the Cityscapes and $C = 4$ for the Edges2Shoes dataset (since edge maps are greyscale). We make either 100, 1000, or all training samples available in paired form, to investigate whether FactorGAN can improve upon GAN by exploiting the remaining unpaired samples or match its quality if there are none.

For evaluation, we randomly assign $80\%$ of validation data to a "test-train" and the rest to a "test-test" partition. We train an LSGAN discriminator (Mao et al., 2017) with the architecture shown in Table 5 (but half the filters in each layer) on the test-train partition for 40 epochs to distinguish real from generated samples, before measuring its loss on the test set. We continuously sample from the generator during training and testing instead of using a fixed set of samples to better approximate the true generator distribution. As evaluation metric, we use the average test loss over 10 training runs, which was shown to correlate with subjective ratings of visual quality (Im et al., 2018) and also with our own quality judgements throughout this study. A larger value indicates better performance, as we use a flipped sign compared to Im et al. (2018). While the quantitative results appear indicative of output quality, accurate GAN evaluation is still an open problem and so we encourage the reader to judge generated examples given in Section A.5.

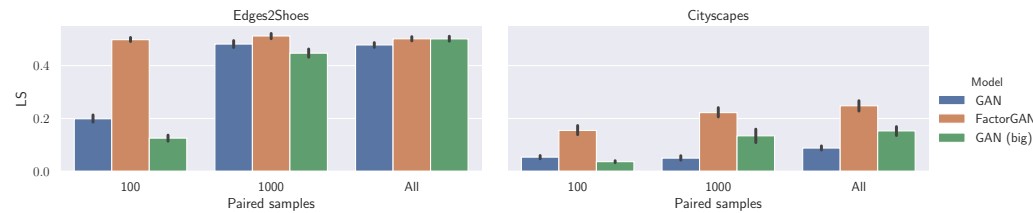

Figure 2: GAN and FactorGAN output quality estimated by the LS metric for different datasets and numbers of paired samples. Error bars show 95% confidence intervals.

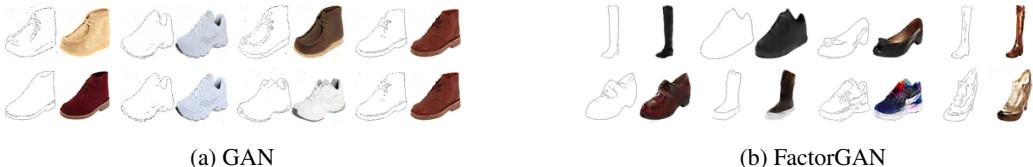

(a) GAN           (b) FactorGAN

Figure 3: Examples generated for the Edges2Shoes dataset using 100 paired samples

**Results** Our FactorGAN achieves better or similar output quality compared to the GAN baseline in all cases, as seen in Figure 2. For the Edges2Shoes dataset, the performance gains are most pronounced for small numbers of paired samples. On the more complex Cityscapes dataset, FactorGAN outperforms GAN by a large margin independent of training set size, even when the discriminators are closely matched in size. This suggests that FactorGAN converges with fewer training iterations for $G_\phi$, although the exact cause is unclear and should be investigated in future work.

We show some generated examples in Figure 3. Due to the small number of available paired samples, we observe a strong mode collapse of the GAN in Figure 3a, while FactorGAN provides high-fidelity, diverse outputs, as shown in Figure 3b. Similar observations can be made for the Cityscapes dataset when using 100 paired samples (see Section A.5.2).

### 4.3 IMAGE SEGMENTATION

Our approach extends to the case of conditional generation (see Section 2.3), so we tackle a complex and important image segmentation task on the Cityscapes dataset, where we ask the generator to predict a segmentation map for a city scene (instead of generating both from scratch as in Section 4.2).

**Experimental setup** We downsample the scenes and segmentation maps to $128 \times 128$ pixels and use a U-Net architecture (Ronneberger et al., 2015) (shown in Table 6 with $W = 7$ and $C = 3$) as segmentation model. For FactorGAN, we use one marginal discriminator to match the distribution of real and fake segmentation maps to ensure realistic predictions, which enables training with isolated city scenes and segmentation maps. To ensure the correct predictions for each city scene, a p- and a q-dependency discriminator learns the input-output relationship using joint samples, both employing a convolutional architecture shown in Table 5. Note that as in Section 4.2, we output segmentation maps in the RGB space instead of performing classification. In addition to the MSE in the RGB space, we compute the widely used pixel-wise classification accuracy (Cordts et al., 2016) by assigning each output pixel to the class whose colour has the lowest Euclidean distance in RGB space.

Using the same experimental setup (including network architectures), we also implement the Cycle-GAN (Zhu et al., 2017) as an unsupervised baseline. For the CycleGAN objective, the same GAN losses as shown in (1) and (2) are used[5].

**Results** The results in Figure 4 demonstrate that our approach can exploit additional unpaired samples to deliver better MSE and accuracy than a GAN and less noisy outputs as seen in Figure 5.

---

[5]Code to perform one training iteration and default loss weights taken from the official codebase at
`https://github.com/junyanz/pytorch-CycleGAN-and-pix2pix`

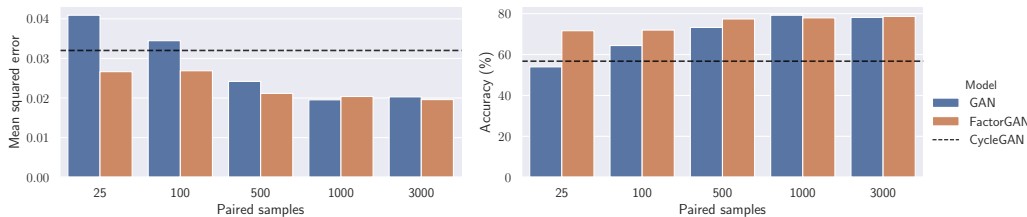

Figure 4: MSE (left) and accuracy (right) obtained on the Cityscapes dataset with different numbers of paired training samples for the GAN and FactorGAN

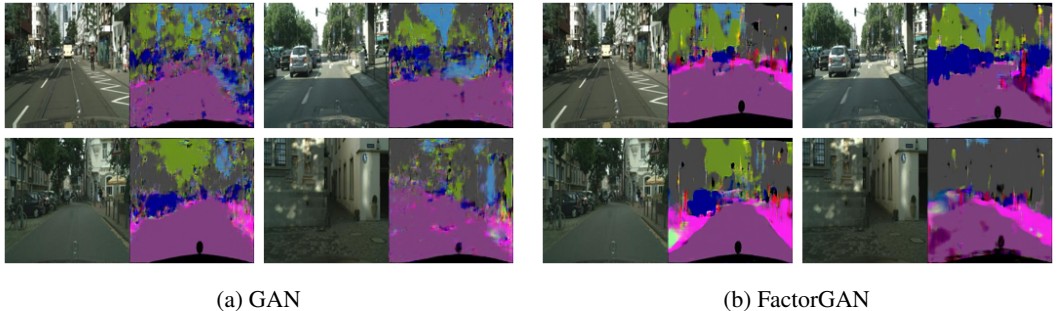

(a) GAN                                                    (b) FactorGAN

Figure 5: Segmentation predictions made on the Cityscapes dataset for the same set of test inputs, compared between models, using 100 paired samples for training

When using only 25 paired samples, FactorGAN reaches 71.6% accuracy, outperforming both GAN and CycleGAN by an absolute 17.7% and 14.9%, respectively. CycleGAN performs better than GAN only in this setting, and increasingly falls behind both GAN and FactorGAN with a growing number of paired samples, likely since GAN and FactorGAN are able to improve their input-output mapping gradually while CycleGAN remains reliant on its cycle consistency assumption. These findings suggest that FactorGAN can efficiently learn the dependency structure from few paired samples with more accuracy than a CycleGAN that is limited by its simplistic cycle consistency assumption.

## 4.4 AUDIO SOURCE SEPARATION

We apply our method to audio source separation as another conditional generation task to investigate whether it transfers across domains. Specifically, we separate music signals into singing voice and accompaniment, as detailed in Section A.2. As in Section 4.3, we find that FactorGAN provides better separtion than GAN, suggesting that our factorisation is useful across problem domains.

## 5 DISCUSSION

We find that FactorGAN outperforms GAN across all experiments when additional incomplete samples are available, especially when they are abundant in comparison to the number of joint samples. When using only joint observations, FactorGAN should be expected to match the GAN in quality, and it does so quite closely in most of our experiments. Surprisingly, it outperforms GAN in some scenarios such as image segmentation even with matched discriminator sizes – a phenomenon we do not fully understand yet and should be investigated in the future. For image segmentation, FactorGAN substantially improves segmentation accuracy compared to the fully unsupervised CycleGAN model even when only using 25 paired examples, indicating that it can efficiently exploit the pairing information.

Since the p-dependency discriminator does not rely on generator samples that change during training, it could be pre-trained to reduce computation time, but this led to sudden training instabilities in our experiments. We suspect that this is due to a mismatch between training and testing conditions for the p-dependency discriminator since it is trained on real but evaluated on fake data, and neural networks

can yield overly confident predictions outside the support of the training set (Gal & Ghahramani, 2016). Therefore, we expect classifiers with better uncertainty calibration to alleviate this issue.

# 6 CONCLUSION

In this paper, we demonstrated how a joint distribution can be factorised into a set of marginals and dependencies, giving rise to the FactorGAN – a GAN in which the discriminator is split into parts that can be independently trained with incomplete observations. For both generation and conditional prediction tasks in multiple domains, we find that FactorGAN outperforms the standard GAN when additional incomplete observations are available. For Cityscapes scene segmentation in particular, FactorGAN achieves a much higher accuracy than the supervised GAN as well as the unsupervised CycleGAN, while requiring only 25 of all examples to be annotated.

Factorising discriminators enables incorporating more prior knowledge into the design of neural architectures in GANs, which could improve empirical results in applied domains. The presented factorisation is generally applicable independent of model choice, so it can be readily integrated into many existing GAN-based approaches. Since the joint density can be factorised in different ways, multiple extensions are conceivable depending on the particular application (as shown in Section A.3). This paper derives FactorGAN from the original GAN proposed by Goodfellow et al. (2014) by exploiting the probabilistic view of the optimal discriminator. Adapting the FactorGAN to alternative GAN objectives (such as the Wasserstein GAN (Arjovsky et al., 2017)) might be possible as well. Instead of relying on additional techniques such as spectral normalisation to ensure training stability, which our theory does not explicitly incorporate, this would enable the use of an inherently more stable GAN variant with the same theoretical guarantees.

ACKNOWLEDGMENTS

We thank Emmanouil Benetos for his helpful feedback. Daniel Stoller is funded by EPSRC grant EP/L01632X/1.

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

# A  APPENDIX

## A.1  TABLES

Table 1: The architecture of our generator on the MNIST dataset. All layers have biases.

| Layer | Input shape | Outputs | Output shape | Activation |
|---|---|---|---|---|
| FC | 50 | 128 | 128 | ReLU |
| FC | 128 | 128 | 128 | ReLU |
| FC | 128 | 1568 | $56 \times 28 \times 1$ | Sigmoid |

Table 2: The architecture of our discriminators on the paired MNIST dataset. $W = 28$ for marginal, $W = 56$ for dependency discriminators.

| Layer | Input shape | Outputs | Output shape | Activation |
|---|---|---|---|---|
| FC | $W \cdot 28$ | 128 | 128 | LeakyReLU |
| FC | 128 | 128 | 128 | LeakyReLU |
| FC | 128 | 1 | 1 | - |

Table 3: The architecture of our MNIST classifier. Dropout with probability $0.5$ is applied to FC1 outputs.

| Layer | Input shape | Filter size | Stride | Outputs | Output shape | Activation |
|---|---|---|---|---|---|---|
| Conv | $28 \times 28 \times 1$ | $5 \times 5$ | $1 \times 1$ | 10 | $28 \times 28 \times 10$ | - |
| AvgPool | $28 \times 28 \times 10$ | $2 \times 2$ | $2 \times 2$ | 10 | $12 \times 12 \times 10$ | LeakyReLU |
| Conv | $12 \times 12 \times 10$ | $5 \times 5$ | $1 \times 1$ | 20 | $12 \times 12 \times 20$ | - |
| AvgPool | $12 \times 12 \times 20$ | $2 \times 2$ | $2 \times 2$ | 20 | $4 \times 4 \times 20$ | LeakyReLU |
| FC1 | 320 | - | - | 50 | 50 | LeakyReLU |
| FC2 | 50 | - | - | 10 | 10 | - |

Table 4: The architecture of our convolutional generator. "ConvT" represent transposed convolutions. All layers have biases. The number of output channels $C$ depends on the task.

| Layer | Input shape | Filter size | Stride | Outputs | Output shape | Activation |
|---|---|---|---|---|---|---|
| ConvT | $1 \times 1 \times 50$ | $4 \times 4$ | $1 \times 1$ | 1024 | $4 \times 4 \times 1024$ | ReLU |
| ConvT | $4 \times 4 \times 1024$ | $4 \times 4$ | $2 \times 2$ | 512 | $8 \times 8 \times 512$ | ReLU |
| ConvT | $8 \times 8 \times 512$ | $4 \times 4$ | $2 \times 2$ | 256 | $16 \times 16 \times 256$ | ReLU |
| ConvT | $16 \times 16 \times 256$ | $4 \times 4$ | $2 \times 2$ | 128 | $32 \times 32 \times 128$ | ReLU |
| ConvT | $32 \times 32 \times 128$ | $4 \times 4$ | $2 \times 2$ | 64 | $64 \times 64 \times 64$ | ReLU |
| Conv | $64 \times 64 \times 64$ | $4 \times 4$ | $1 \times 1$ | $C$ | $64 \times 64 \times C$ | Sigmoid |

Table 5: The architecture of our convolutional discriminator. All layers except FC have biases. $W$, $H$ and $C$ are set for each task so that the dimensions of the input data are matched.

| Layer | Input shape | Filter size | Stride | Outputs | Output shape | Activation |
|---|---|---|---|---|---|---|
| Conv | $2^W \times 2^H \times C$ | $4 \times 4$ | $2 \times 2$ | 32 | $2^{W-1} \times 2^{H-1} \times 32$ | LeakyReLU |
| Conv | $2^{W-1} \times 2^{H-1} \times 32$ | $4 \times 4$ | $2 \times 2$ | 64 | $2^{W-2} \times 2^{H-2} \times 64$ | LeakyReLU |
| Conv | $2^{W-2} \times 2^{H-2} \times 64$ | $4 \times 4$ | $2 \times 2$ | 128 | $2^{W-3} \times 2^{H-3} \times 128$ | LeakyReLU |
| Conv | $2^{W-3} \times 2^{H-3} \times 128$ | $4 \times 4$ | $2 \times 2$ | 256 | $2^{W-4} \times 2^{H-4} \times 256$ | LeakyReLU |
| Conv | $2^{W-4} \times 2^{H-4} \times 256$ | $4 \times 4$ | $2 \times 2$ | 512 | $2^{W-5} \times 2^{H-5} \times 512$ | LeakyReLU |
| FC | $2^{W-5} \cdot 2^{H-5} \cdot 512$ | - | - | 1 | 1 | LeakyReLU |

Table 6: The architecture of our U-Net. The height $H$ and number of input channels $C$ depends on the experiment. MP is maxpooling with stride 2. FC has noise as input. UpConv performs transposed convolution with stride 2. Concat concatenates the current feature map with one from the downstream path. The final output is computed depending on the task (see text for more details)

| Layer | Input (shape) | Outputs | Output shape |
|---|---|---|---|
| DoubleConv1 | $2^W \times 128 \times C$ | 32 | $2^W \times 128 \times 32$ |
| MP1 | $2^W \times 128 \times 32$ | 32 | $2^{W-1} \times 64 \times 32$ |
| DoubleConv2 | $2^{W-1} \times 64 \times 32$ | 64 | $2^{W-1} \times 64 \times 64$ |
| MP2 | $2^{W-1} \times 64 \times 64$ | 64 | $2^{W-2} \times 32 \times 64$ |
| DoubleConv3 | $2^{W-2} \times 32 \times 64$ | 64 | $2^{W-2} \times 32 \times 128$ |
| MP3 | $2^{W-2} \times 32 \times 128$ | 128 | $2^{W-3} \times 16 \times 128$ |
| DoubleConv4 | $2^{W-3} \times 16 \times 128$ | 256 | $2^{W-3} \times 16 \times 256$ |
| MP4 | $2^{W-3} \times 16 \times 256$ | 256 | $2^{W-4} \times 8 \times 256$ |
| DoubleConv5 | $2^{W-4} \times 8 \times 256$ | 256 | $2^{W-4} \times 8 \times 256$ |
| FC | 50 | $2^{W-4} \cdot 16$ | $2^{W-4} \times 8 \times 2$ |
| Concat | DoubleConv5 | - | $2^{W-4} \times 8 \times 258$ |
| UpConv | $2^{W-4} \times 8 \times 258$ | 256 | $2^{W-3} \times 16 \times 258$ |
| Concat | DoubleConv4 | 514 | $2^{W-3} \times 16 \times 514$ |
| Conv | $2^{W-3} \times 16 \times 514$ | 128 | $2^{W-3} \times 16 \times 128$ |
| UpConv | $2^{W-3} \times 16 \times 128$ | 128 | $2^{W-2} \times 32 \times 128$ |
| Concat | DoubleConv3 | 256 | $2^{W-2} \times 32 \times 256$ |
| Conv | $2^{W-2} \times 32 \times 256$ | 64 | $2^{W-2} \times 32 \times 64$ |
| UpConv | $2^{W-2} \times 32 \times 64$ | 64 | $2^{W-1} \times 64 \times 64$ |
| Concat | DoubleConv2 | 128 | $2^{W-1} \times 64 \times 128$ |
| Conv | $2^{W-1} \times 64 \times 128$ | 32 | $2^{W-1} \times 64 \times 32$ |
| UpConv | $2^{W-1} \times 64 \times 32$ | 32 | $2^W \times 128 \times 32$ |
| Concat | DoubleConv1 | 64 | $2^W \times 128 \times 64$ |
| Conv | $2^W \times 128 \times 64$ | 32 | $2^W \times 128 \times 32$ |
| Conv | $2^W \times 128 \times 32$ | $C$ | $2^W \times 128 \times C$ |

Table 7: The DoubleConv neural network block used in the U-Net. Conv uses a $3 \times 3$ filter size.

| Layer | Input shape | Outputs | Output shape |
|---|---|---|---|
| Conv | $W \times H \times C$ | $\frac{C}{2}$ | $W \times H \times \frac{C}{2}$ |
| BatchNorm & ReLU | $W \times H \times \frac{C}{2}$ | - | $W \times H \times \frac{C}{2}$ |
| Conv | $W \times H \times \frac{C}{2}$ | $\frac{C}{2}$ | $W \times H \times \frac{C}{2}$ |
| BatchNorm & ReLU | $W \times H \times \frac{C}{2}$ | - | $W \times H \times \frac{C}{2}$ |

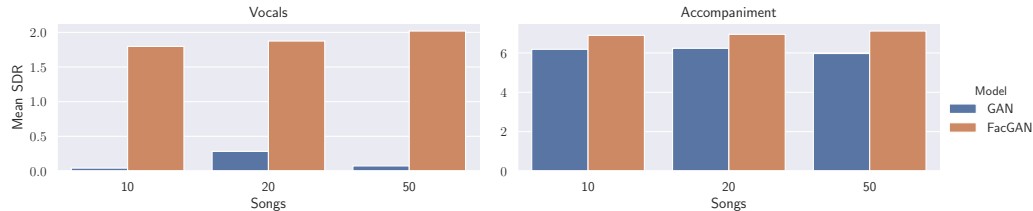

Figure 6: GAN and FactorGAN separation performance for different numbers of paired samples

## A.2 AUDIO SOURCE SEPARATION EXPERIMENT

For our audio source separation experiment, our generator $G_\phi$ takes a music spectrogram $\mathbf{m}$ along with noise $\mathbf{z}$ and maps it to an estimate of the accompaniment and vocal spectra $\mathbf{a}$ and $\mathbf{v}$, implicitly defining an output probability $q_\phi(\mathbf{a}, \mathbf{v}|\mathbf{m})$. We define the joint real and generated distributions that should be matched as $p(\mathbf{m}, \mathbf{a}, \mathbf{v})$ and $q(\mathbf{m}, \mathbf{a}, \mathbf{v}) = q_\phi(\mathbf{a}, \mathbf{v}|\mathbf{m})p(\mathbf{m})$. Since the source signals in our dataset are simply added in the time-domain to produce the mixture, this approximately applies to the spectrogram as well, so we assume that $p(\mathbf{m}|\mathbf{a}, \mathbf{v}) = \delta(\mathbf{m} - \mathbf{a} - \mathbf{v})$. We can constrain our generator $G_\phi$ to make predictions that always satisfy this condition, thereby taking care of the input-output relationship manually, similarly to Sønderby et al. (2017). Instead of predicting the sources directly, a mask $\mathbf{b}$ with values in the range $[0, 1]$ is computed, and the accompaniment and vocals are estimated as $\mathbf{b} \odot \mathbf{m}$ and $(\mathbf{b} - 1) \odot \mathbf{m}$, respectively. As a result, $q(\mathbf{m}|\mathbf{a}, \mathbf{v}) = p(\mathbf{m}|\mathbf{a}, \mathbf{v})$, so we can simplify the joint density ratio to

$$\frac{p(\mathbf{m}, \mathbf{a}, \mathbf{v})}{q(\mathbf{m}, \mathbf{a}, \mathbf{v})} = \frac{p(\mathbf{a}, \mathbf{v})p(\mathbf{m}|\mathbf{a}, \mathbf{v})}{q(\mathbf{a}, \mathbf{v})q(\mathbf{m}|\mathbf{a}, \mathbf{v})} = \frac{p(\mathbf{a}, \mathbf{v})}{q(\mathbf{a}, \mathbf{v})} = \frac{c_P(\mathbf{a}, \mathbf{v})}{c_Q(\mathbf{a}, \mathbf{v})} \frac{p(\mathbf{a})}{q(\mathbf{a})} \frac{p(\mathbf{v})}{q(\mathbf{v})}, \tag{6}$$

meaning that the discriminator(s) in the GAN and the FactorGAN only require $(\mathbf{a}, \mathbf{v})$ pairs, but not the mixture $\mathbf{m}$ as additional input, as the correct input-output relationship is already incorporated into the generator. Furthermore, the last equality suggests a FactorGAN application with one marginal discriminator for each source along with dependency discriminators to model source dependencies.

**Dataset**  We use MUSDB (Rafii et al., 2017) as multi-track dataset for our experiment, featuring 100 songs for training and 50 songs for testing. Each song is downsampled to 22.05 KHz before spectrogram magnitudes are computed, using an STFT with a 512-sample window and a 256-sample hop[6]. Snippets with 128 timeframes each are created by cropping each song's full spectrogram at regular intervals of 64 timeframes. Thus, the generator only separates snippets $\mathbf{m} \in \mathbb{R}_{\geq 0}^{256 \times 128}$ and outputs predictions of the same shape, however this does not change the derivation presented in Equation (6), and longer inputs at test time can be processed by partitioning them into snippets and concatenating the model predictions.

**Experimental setup**  For our generator, we use the U-Net architecture detailed in Table 6 with $W = 8$ and $C = 1$. We use the convolutional discriminator described in Table 5 with $W = 8$, $H = 7$ and $C = 1$. The source dependency discriminators take two sources as input via concatenation along the channel dimension, so they use $C = 2$.

In each experiment, we vary the number of training songs whose snippets are available for paired training between 10, 20 and 50 and compare between GAN and FactorGAN. The spectrograms predicted on the test set are converted to audio with the inverse STFT by reusing the phase from the mixture, and then evaluated using the signal-to-distortion ratio (SDR), a well-established evaluation metric for source separation (Vincent et al., 2006).

**Results**  Figure 6 shows our separation results. Compared to a GAN, the separation performance is significantly higher using FactorGAN. As expected, FactorGAN improves slightly with more paired examples, which is not the case for the GAN – here we find that the vocal output becomes too quiet

---

[6]This results in 257 frequency bins but we discard the bin with the highest frequency to obtain a power of 2 and thus avoid padding issues in our network architectures.

when increasing the number of songs for training, possibly a sign of mode collapse. Similarly to the results seen in the image pair generation experiments, we suspect that the FactorGAN discriminator might approximate the joint density $\tilde{D}(\mathbf{x})$ more closely than the GAN discriminator due to its use of multiple discriminators, although the reasons for this are not yet understood.

### A.3 POSSIBLE EXTENSIONS

We can decompose the joint density ratio $\frac{p_x(\mathbf{x})}{q_x(\mathbf{x})}$ in other ways than shown in Equation 3 in the paper. In the following, we discuss two additional possibilities.

#### A.3.1 HIERARCHICAL FACTORGAN

The decomposition of the joint density ratio could be applied recursively, splitting the obtained marginals further into "sub-marginals" and their dependencies, which could be repeated multiple times. In addition to training with incomplete observations where only a single part is given, this also allows making use of samples where only sub-parts of these parts are given and is thus more flexible than a single factorisation as used in the standard FactorGAN.

As a demonstration, we split each marginal $\mathbf{x}^i$ further into a group of $J_i$ marginals, $J_i \leq |\mathcal{D}_i|$, and their dependencies, without further recursion for simplicity:

$$\frac{p_x(\mathbf{x})}{q_x(\mathbf{x})} = \frac{c_P(\mathbf{x})}{c_Q(\mathbf{x})} \prod_{i=1}^{K} \frac{p_x^i(\mathbf{x}^i)}{q_x^i(\mathbf{x}^i)} = \frac{c_P(\mathbf{x})}{c_Q(\mathbf{x})} \left[ \prod_{i=1}^{K} \frac{c_P^i(\mathbf{x}^i)}{c_Q^i(\mathbf{x}^i)} \left[ \prod_{j=1}^{J} \frac{p_x^{i,j}(\mathbf{x}^{i,j})}{q_x^{i,j}(\mathbf{x}^{i,j})} \right] \right]. \tag{7}$$

$c_P^i$ and $c_Q^i$ are dependency terms analogously to $c_P$ and $c_Q$, but only defined on marginal variable $\mathbf{x}^i$, whose $J$ "sub-marginals" are denoted by $\mathbf{x}^{i,1}, \ldots, \mathbf{x}^{i,J}$.

Such a hierarchical decomposition might also be beneficial if the data is known to be generated from a hierarchical process. We leave the empirical exploration of this concept to future work.

#### A.3.2 AUTOREGRESSIVE FACTORGAN

For a multi-dimensional variable $\mathbf{x} = [\mathbf{x}^1, \mathbf{x}^2, \ldots, \mathbf{x}^T]$ composed of $T$ elements arranged in a sequence, such as time series data, the joint density ratio can also be decomposed in a causal, auto-regressive fashion:

$$\frac{p_x(\mathbf{x})}{q_x(\mathbf{x})} = \frac{p_x^1(\mathbf{x}^1)}{q_x^1(\mathbf{x}^1)} \prod_{i=2}^{T} \frac{c_P(\mathbf{x}^1, \ldots, \mathbf{x}^i)}{c_Q(\mathbf{x}^1, \ldots, \mathbf{x}^i)} \frac{p_x(\mathbf{x}^i)}{q_x(\mathbf{x}^i)} \tag{8}$$

$$= \frac{p_x^1(\mathbf{x}^1)}{q_x^1(\mathbf{x}^1)} \prod_{i=2}^{T} \frac{p_x(\mathbf{x}_i | \mathbf{x}_1, \ldots, \mathbf{x}_{i-1})}{q_x(\mathbf{x}_i | \mathbf{x}_1, \ldots, \mathbf{x}_{i-1})} \tag{9}$$

Note that $c_P$ is defined here as $\frac{p(\mathbf{x})}{p(\mathbf{x}^1, \ldots, \mathbf{x}^{i-1})p(\mathbf{x}^i)}$ ($c_Q$ analogously using $q_x$). Equation (8) suggests an auto-regressive version of FactorGAN in which the generator output quality at each time-step $i$ is evaluated using a marginal discriminator that estimates $\frac{p_x(\mathbf{x}^i)}{q_x(\mathbf{x}^i)}$ combined with dependency discriminators that model the dependency between the current and all past time-steps.

The final product formulation in Equation (9) reveals a close similarity to auto-regressive models and suggests a modification of the normal GAN with an auto-regressive discriminator that rates an input at each time-step given the previous ones. Using a derivation analogous to the one shown in Section A.4, this implies taking the unnormalised discriminator outputs at each time-step, summing them, and applying a sigmoid non-linearity to obtain the overall estimate of the probability $\tilde{D}(\mathbf{x})$. A similar implementation was used before in Mogren (2016), attempting to stabilise GAN training with recurrent neural networks as discriminators, but for the first time, we provide a rigorous theoretical justification for this practice here.

### A.4 DISCRIMINATOR COMBINATION

**Definition A.1.** Sigmoid discriminator output. Let $D_{\theta_i}(\mathbf{x}^i) := \sigma(d_{\theta_i}(\mathbf{x}^i)), d_{\theta_i} : \mathbb{R}^{|\mathcal{D}_i|} \to \mathbb{R}$ for all $i \in \{1, \ldots, K\}$, analogously define $D_{\theta_P}^P(\mathbf{x})$ and $D_{\theta_Q}^Q(\mathbf{x})$.

**Definition A.2.** Combined discriminator. Let $D^C(\mathbf{x}) := \sigma(d^P_{\theta_P}(\mathbf{x}) - d^Q_{\theta_Q}(\mathbf{x}) + \sum_{i=1}^K d_{\theta_i}(\mathbf{x}^i))$ be the output of the combined discriminator that is used for training $G_\phi$ using Equation 2.

**Theorem 1.** Combined discriminator approximates $\tilde{D}(\mathbf{x})$. Under definitions A.1 and A.2 and assuming optimally trained sub-discriminators, $D^C(\mathbf{x}) = \tilde{D}(\mathbf{x}) = \frac{p_x(\mathbf{x})}{p_x(\mathbf{x}) + q_x(\mathbf{x})}$.

*Proof.* Proof of Theorem 1 using Definitions A.1 and A.2:

$$
\begin{aligned}
&D^C(\mathbf{x}) \\
&= \sigma\left(d^P_{\theta_P}(\mathbf{x}) - d^Q_{\theta_Q}(\mathbf{x}) + \sum_{i=1}^K d_{\theta_i}(\mathbf{x}^i)\right) \\
&= \left(1 + e^{-d^P_{\theta_P}(\mathbf{x})} e^{d^Q_{\theta_Q}(\mathbf{x})} \prod_{i=1}^K e^{-d_{\theta_i}(\mathbf{x}^i)}\right)^{-1} \\
&= \left(1 + \frac{1 - D^P_{\theta_P}(\mathbf{x})}{D^P_{\theta_P}(\mathbf{x})} \frac{D^Q_{\theta_Q}(\mathbf{x})}{1 - D^Q_{\theta_Q}(\mathbf{x})} \prod_{i=1}^K \frac{1 - D_{\theta_i}(\mathbf{x}^i)}{D_{\theta_i}(\mathbf{x}^i)}\right)^{-1} \\
&= \left(1 + \frac{\prod_{i=1}^K p_x(\mathbf{x}^i)}{p_x(\mathbf{x})} \frac{q_x(\mathbf{x})}{\prod_{i=1}^K q_x^i(\mathbf{x}^i)} \prod_{i=1}^K \frac{q_x^i(\mathbf{x}^i)}{p_x(\mathbf{x}^i)}\right)^{-1} \\
&= \left(1 + \frac{q_x(\mathbf{x})}{p_x(\mathbf{x})}\right)^{-1} \\
&= \frac{p_x(\mathbf{x})}{p_x(\mathbf{x}) + q_x(\mathbf{x})}.
\end{aligned}
\tag{10}
$$

$\square$

## A.5 GENERATED EXAMPLES

### A.5.1 PAIRED MNIST

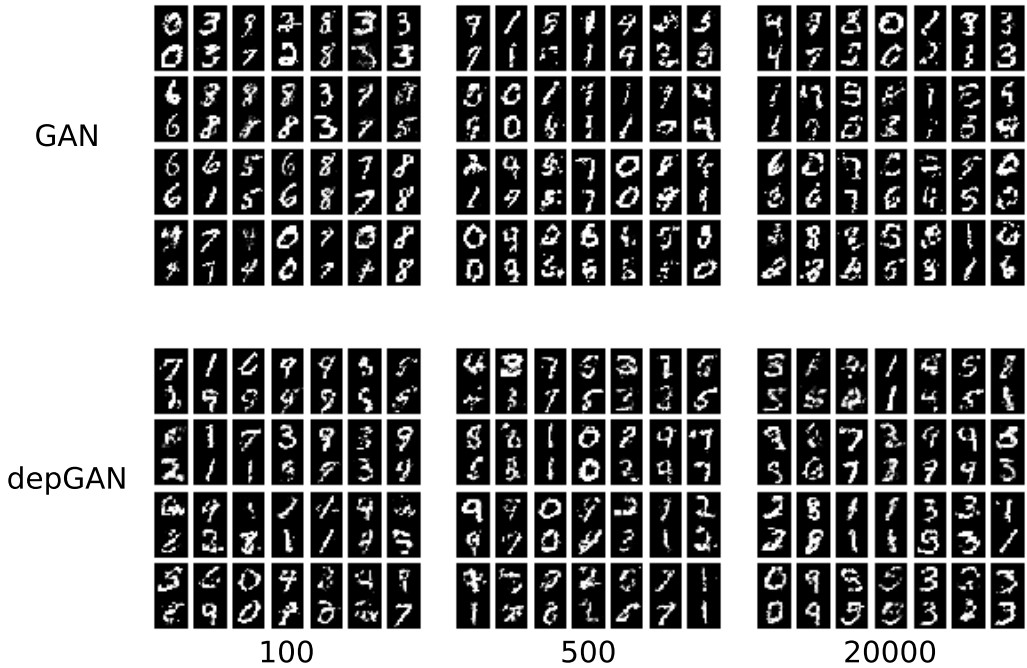

Figure 7: Paired MNIST examples generated by GAN and FactorGAN for different number of paired training samples, using $\lambda = 0.9$.

### A.5.2 IMAGE PAIRS

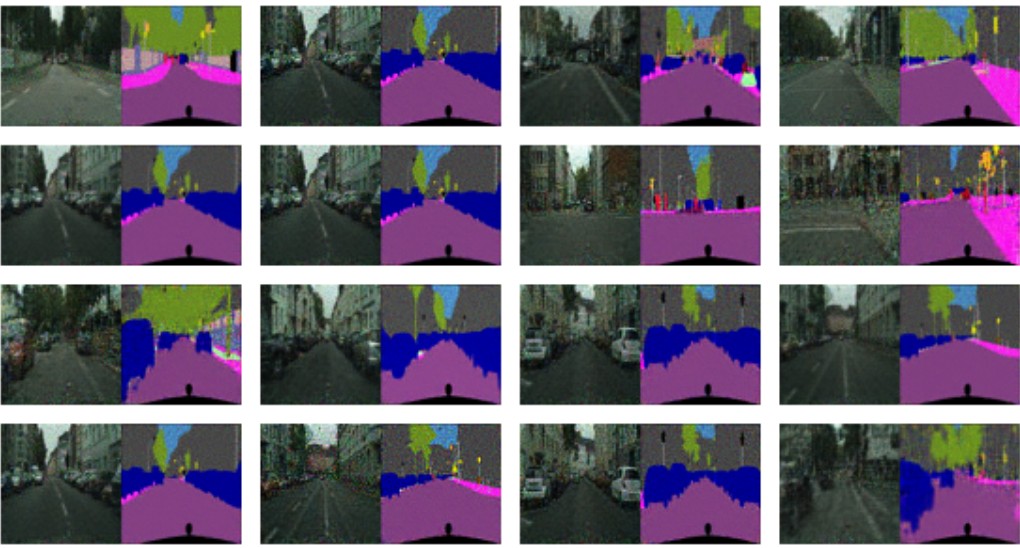

Figure 8: GAN generating image pairs for the Cityscapes dataset using 100 paired samples.

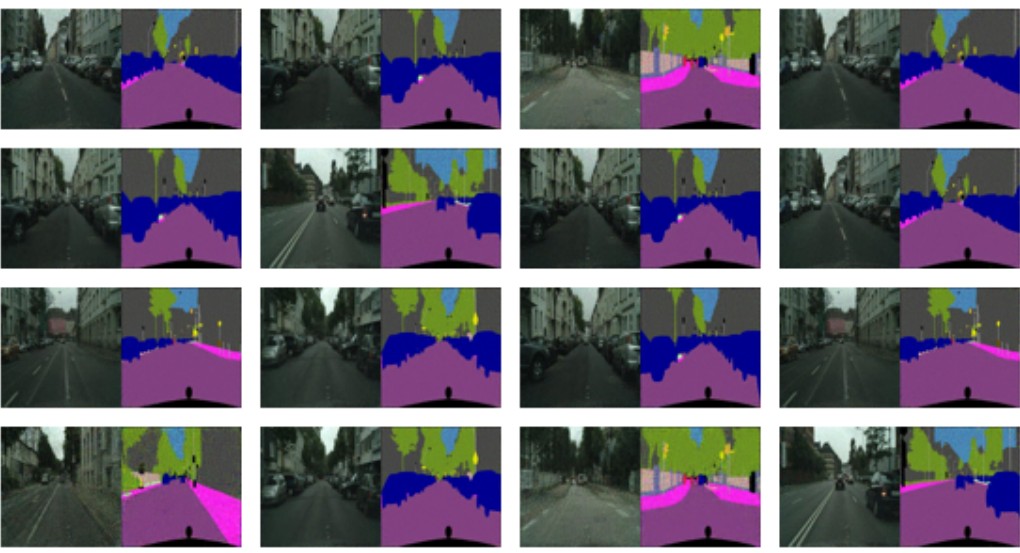

Figure 9: GAN (big) generating image pairs for the Cityscapes dataset using 100 paired samples.

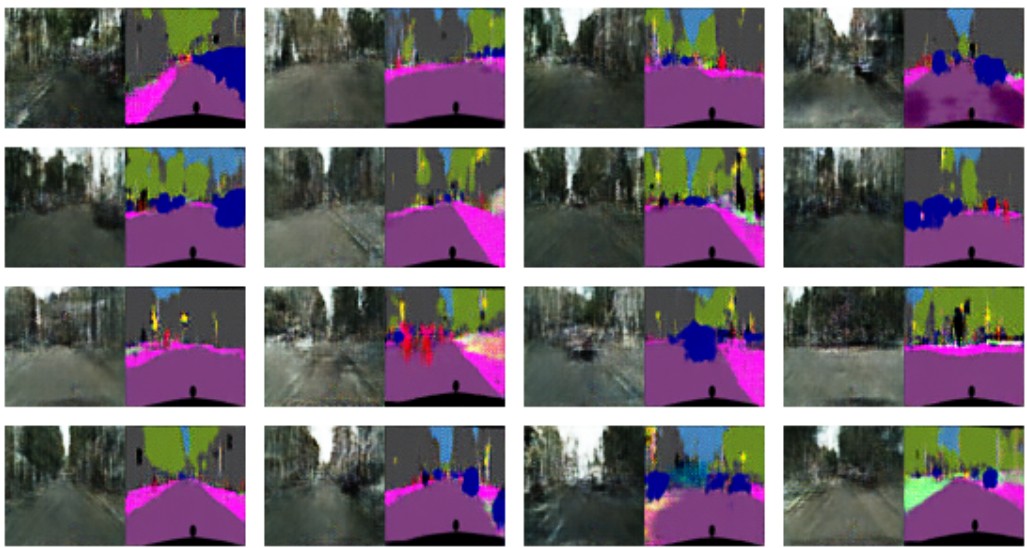

Figure 10: FactorGAN generating image pairs for the Cityscapes dataset using 100 paired samples.

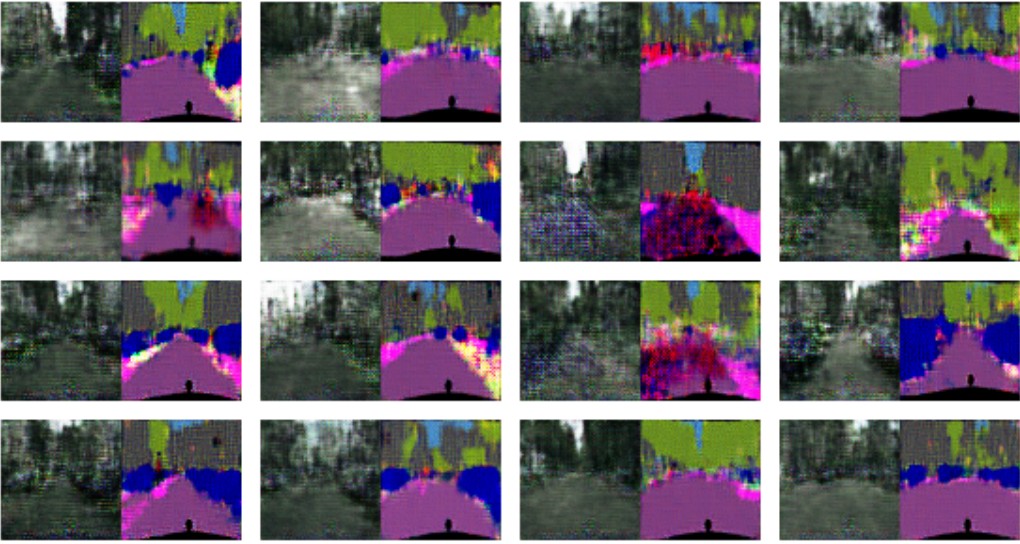

Figure 11: GAN generating image pairs for the Cityscapes dataset using 1000 paired samples.

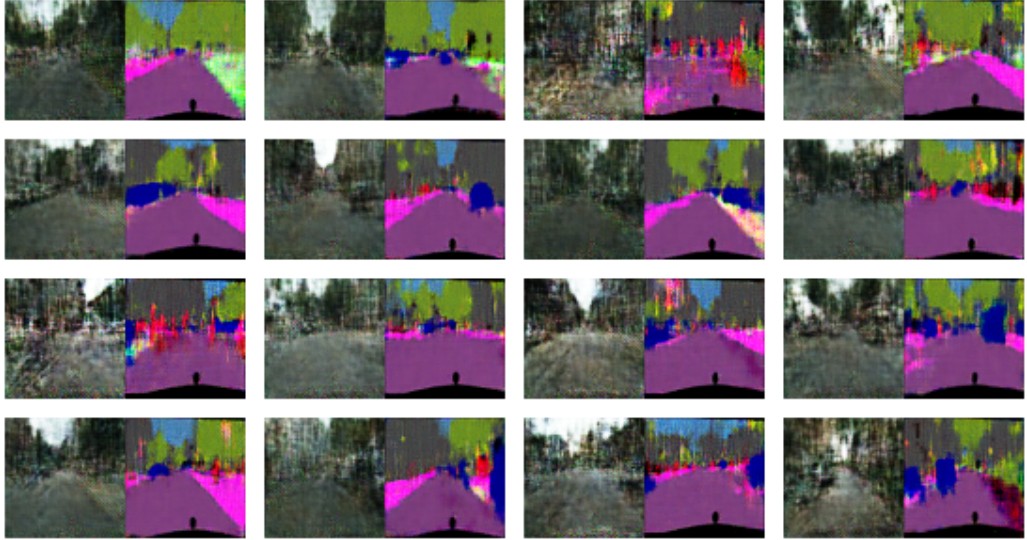

Figure 12: GAN (big) generating image pairs for the Cityscapes dataset using 1000 paired samples.

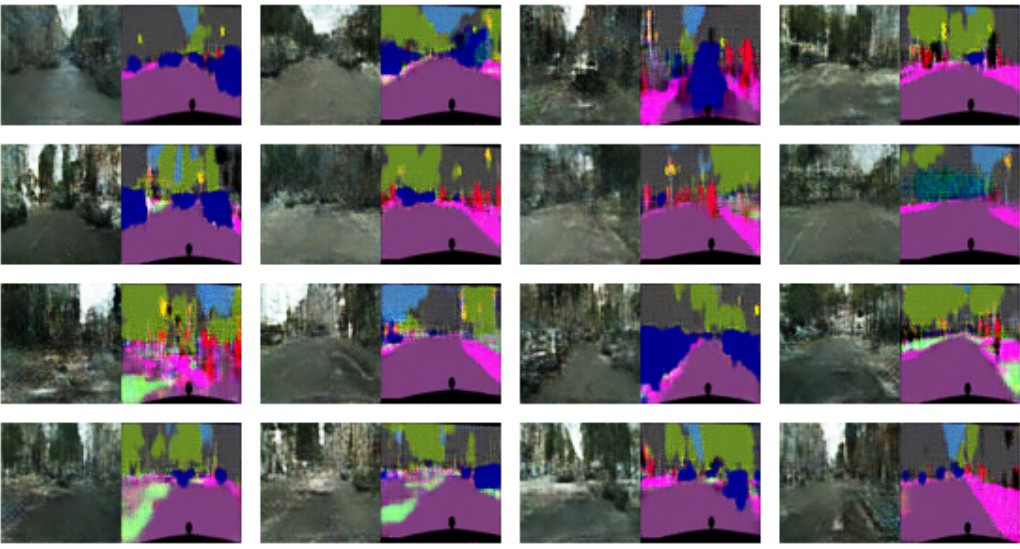

Figure 13: FactorGAN generating image pairs for the Cityscapes dataset using 1000 paired samples.

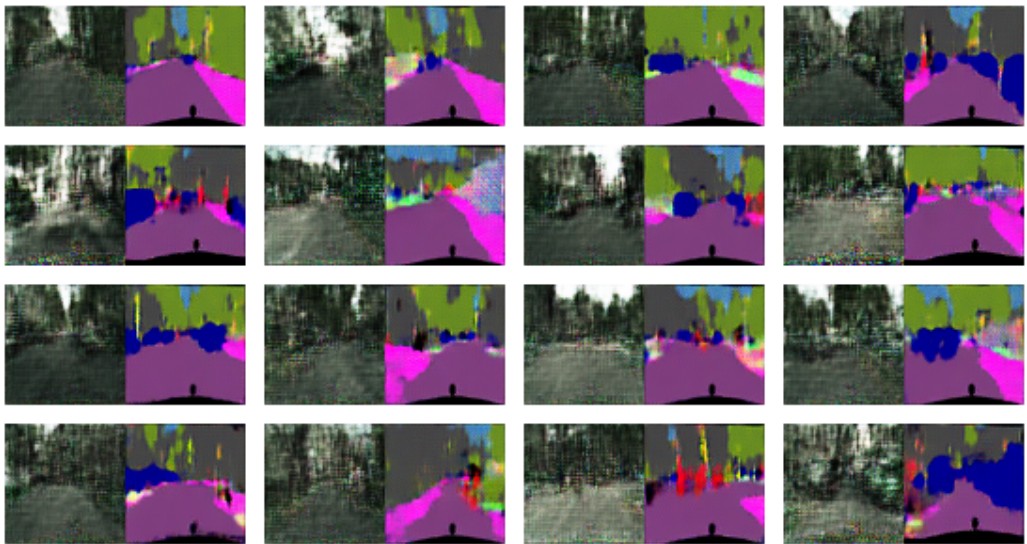

Figure 14: GAN generating image pairs using the full Cityscapes dataset.

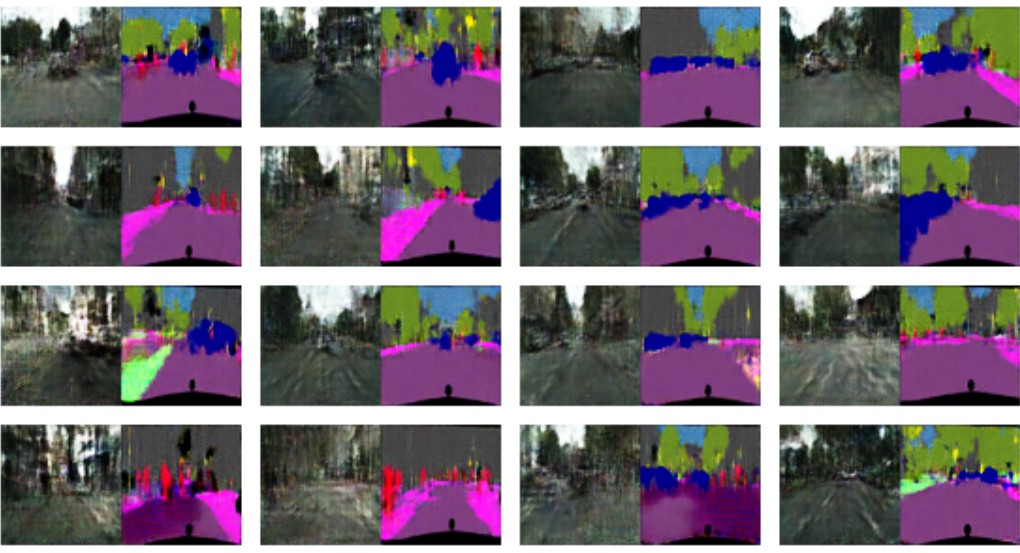

Figure 15: GAN (big) generating image pairs using the full Cityscapes dataset.

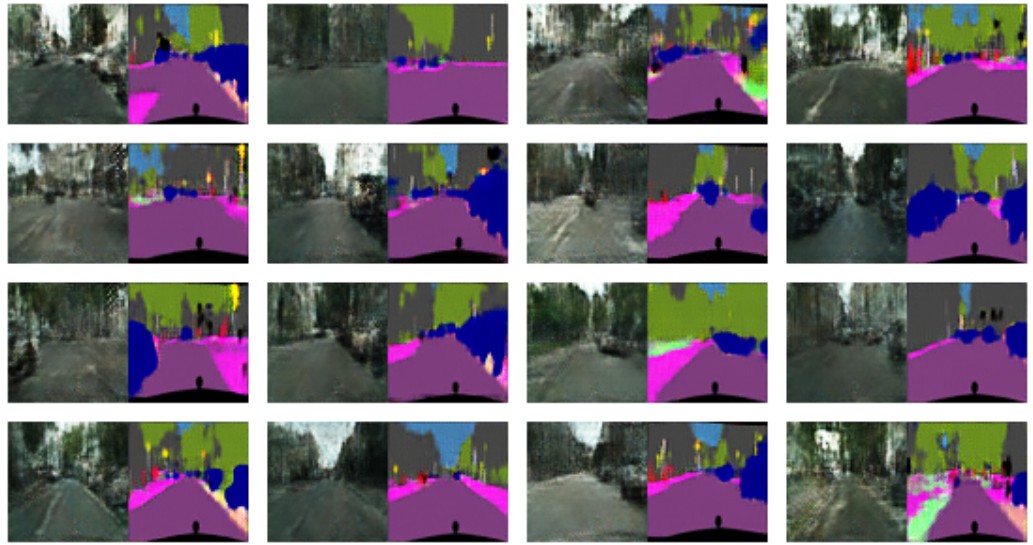

Figure 16: FactorGAN generating image pairs using the full Cityscapes dataset.

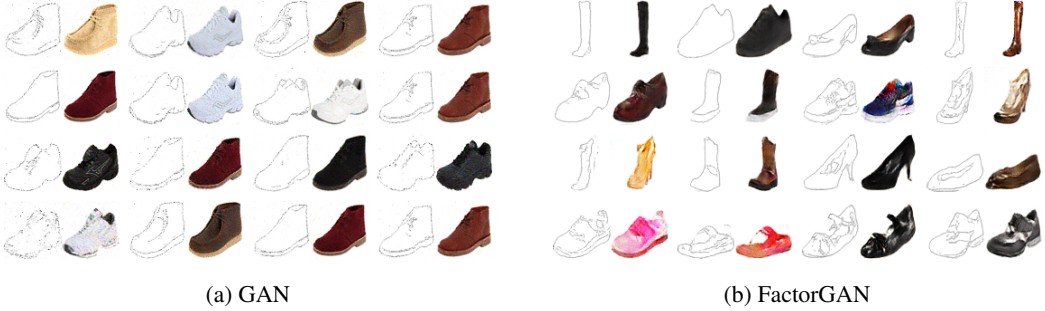

(a) GAN                                    (b) FactorGAN

Figure 17: Image pairs generated for the Edges2Shoes dataset using 100 paired samples.

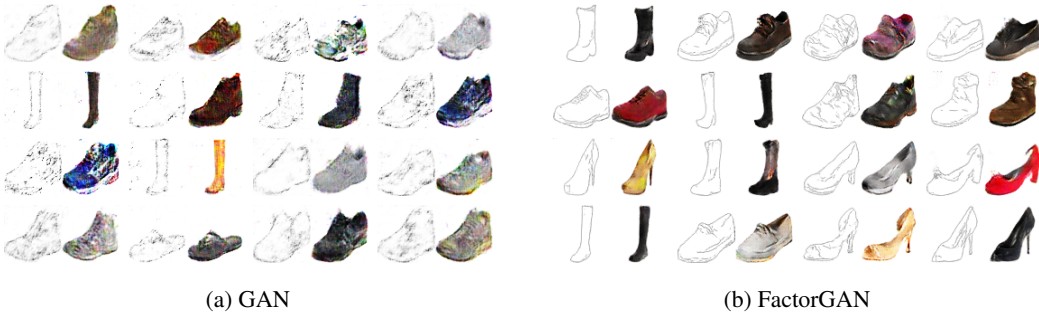

(a) GAN                                    (b) FactorGAN

Figure 18: Image pairs generated for the Edges2Shoes dataset using 1000 paired samples.

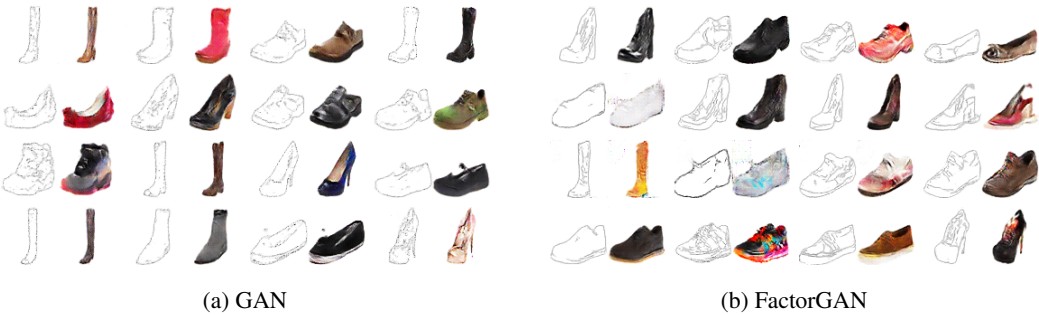

(a) GAN                                    (b) FactorGAN

Figure 19: Image pairs generated for the Edges2Shoes dataset using all samples as paired.

### A.6 IMAGE SEGMENTATION

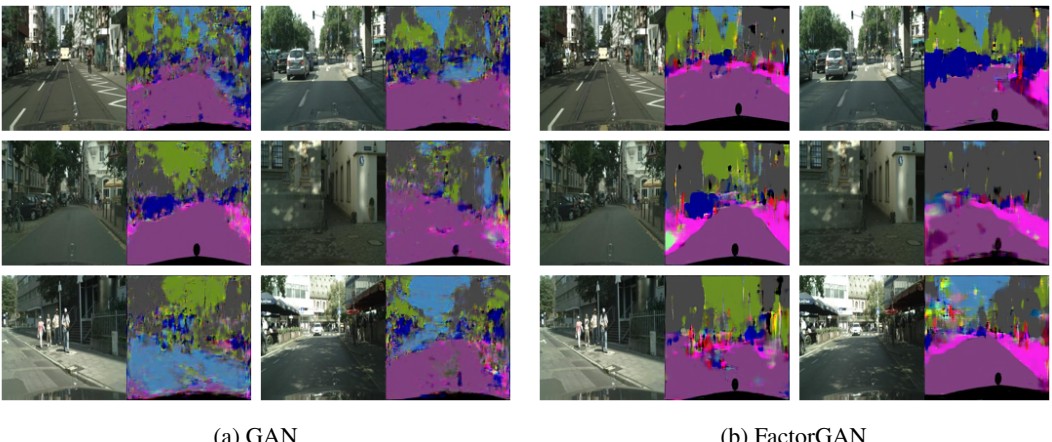

        (a) GAN                           (b) FactorGAN

Figure 20: Segmentation predictions made on the Cityscapes dataset for the same set of test inputs, compared between models, using 100 paired samples for training

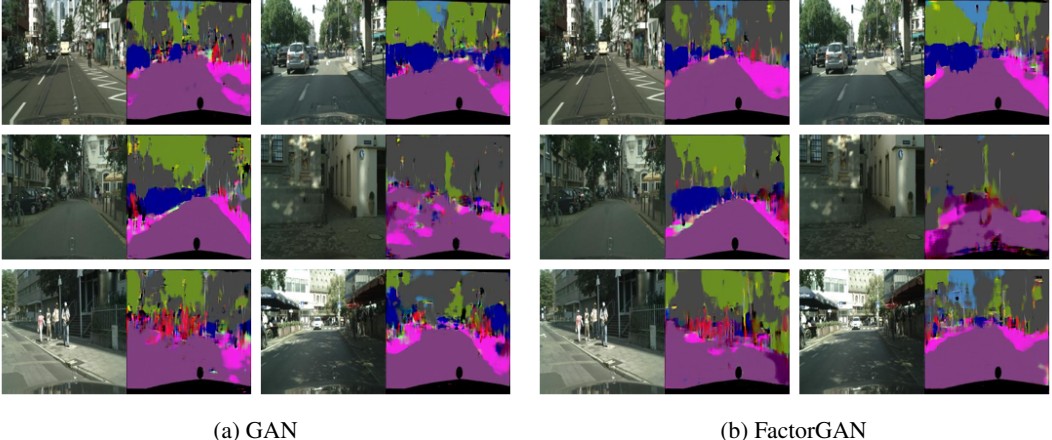

        (a) GAN                           (b) FactorGAN

Figure 21: Segmentation predictions made on the Cityscapes dataset for the same set of test inputs, compared between models, using 1000 paired samples for training

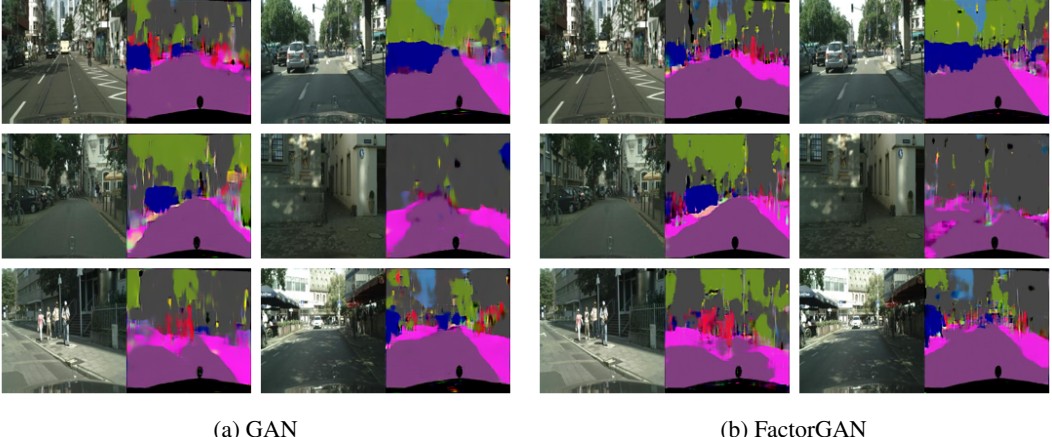

(a) GAN                                        (b) FactorGAN

Figure 22: Segmentation predictions made on the Cityscapes dataset for the same set of test inputs, compared between models, using all paired samples for training

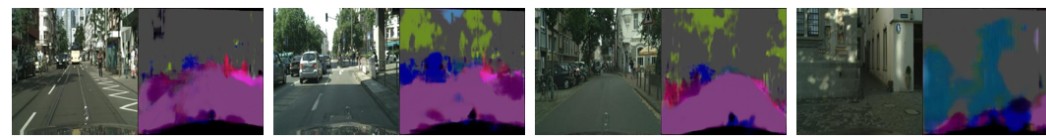

Figure 23: CycleGAN generating image pairs for the Cityscapes dataset without any paired samples.

