# OpenReview forum: "Training Generative Adversarial Networks from Incomplete Observations using Factorised Discriminators"
_ICLR.cc/2020/Conference — Accept (Poster)_

### Official Review · AnonReviewer2 · 2019-10-24
**Official Blind Review #2**

**Rating:** 6

**Review:**

The authors present FactorGANs, which handle missing data scenarios by constructing conditional marginal estimates from ratios of joint and marginal distributions, estimated with GANs. FactorGANs are applied to the problem of semi-supervised (paired+unpaired) translation and demonstrate good performance.

Strengths:
-Nice formulation, which I believe is novel. Well written, good initial results.

Limitations:
-The most serious limitation of the paper is that the technique is not compared with any other semi-supervised methods, such as the Augmented CycleGAN. Because of this it is not clear how the technique compares with SOTA, and so the significance of the paper is not clear.
-The approach scales linearly with the number of marginals, which may limit its applicability to more general imputation tasks.
-The title is the same as an the arxiv paper title, and so the double-blind requirement is trivially violated.

Overall:

A nice formulation, but weak experimental investigations (no comparisons to SOTA semi-supervised translation) make the significance of the paper unclear. This makes it a borderline paper. I strongly encourage the authors to update their experiments accordingly.

Post Response:

Thank you to the authors for the detailed response and additional experimentation. I have updated my rating. It is a nice formulation, and the experimental validation of the technique has been strengthened. The additional experiments (i.e. comparing to the augmented cyclegan) that the authors are following through on will further improve the paper, making it a clear accept.


**Experience Assessment:**

I have read many papers in this area.

**Review Assessment: Checking Correctness Of Derivations And Theory:**

I assessed the sensibility of the derivations and theory.

**Review Assessment: Checking Correctness Of Experiments:**

I assessed the sensibility of the experiments.

**Review Assessment: Thoroughness In Paper Reading:**

I read the paper at least twice and used my best judgement in assessing the paper.

---

> ### Author Response · Authors · 2019-11-14
> **Author response**
>
> Thank you for your feedback on this paper. We hope to clarify some details with the following and thus respond to your questions.
>
> "The most serious limitation of the paper is that the technique is not compared with any other semi-supervised methods, such as the Augmented CycleGAN. Because of this it is not clear how the technique compares with SOTA, and so the significance of the paper is not clear."
>
> Our main contribution is a theoretical foundation for GAN training applicable to generation in a missing data scenario as well as general prediction tasks, and not only limited to image segmentation. While a comparison to SOTA methods for certain sub-tasks (such as image segmentation) could indeed be interesting, our aim was not to claim SOTA for any of these sub-tasks, but to demonstrate that our technique can be applied in a range of possible application scenarios and deliver results in line with our theory (e.g. that performance should increase with more paired samples as the p-dependency discriminator can better estimate its density ratio).
>
> We agree that applying our technique to more tasks and introducing more complex network architectures to reach better performance is certainly worthwhile - given the range of application scenarios we consider and the space constraints for the paper, however, we feel that we have to point to future work in this context.
>
> However, to account for your concerns given the space constraints, we ran additional experiments using the CycleGAN on the image segmentation task (as mentioned also in the response to AnonReviewer4). We used the same network architectures and training setup as the GAN and FactorGAN (so that the standard GAN loss is used alongside spetral normalization). This ensures a fair comparison to GAN and FactorGAN. We included the results in an updated version of the paper, so please refer to the paper for more details. In short, CycleGAN is outperformed by FactorGAN in this setting, even when FactorGAN is only given 25 paired samples, and so FactorGAN is able to model the input-output dependencies more accurately.
>
> We also trained the Augmented CycleGAN by minimally adapting their code [1] to our Cityscapes setting. The only changes were increasing the input resolution from 64x64 to 128x128, and adding one more layer in the discriminator networks due to the higher input resolution. However, the model did not converge, so we are unable to add these results as another baseline.
> Comparison to commonly used missing data imputation methods is also difficult due to the higher number of variables to impute (3 color channels * 128 pixels * 128 pixels per image). We attempted to run missForest [2] but it was too memory-intensive for this reason.
>
> We are currently experimenting with reimplementing the Augmented CycleGAN from scratch, and will update you if we have additional results to share.
>
> "The title is the same as an the arxiv paper title, and so the double-blind requirement is trivially violated."
>
> Please note that we are fully compliant with the ICLR 2020 submission requirements: We fully anonymised both paper and code, and submission on arXiv is explicitly allowed. Citing the call for papers, it says: "However, papers that cite previous related work by the authors and papers that have appeared on non-peered reviewed websites (like arXiv) or that have been presented at workshops (i.e., venues that do not have a publication proceedings) do not violate the policy. The policy is enforced during the whole reviewing process period. Submission of the paper to archival repositories such as arXiv are allowed."
>
> [1] Augmented CycleGAN official codebase. https://github.com/aalmah/augmented_cyclegan
> [2] missForest as implemented in missingPy (https://pypi.org/project/missingpy/)

---

### Official Review · AnonReviewer3 · 2019-10-24
**Official Blind Review #3**

**Rating:** 8

**Review:**

I found this paper very easy and clear to follow - the authors present, what I believe to be an elegant, approach to training a GAN in the presence of missing data or where many marginal samples might be available but very few complete (e.g. paired) samples. The approach proceeds by identifying that the joint distributions (true and approximate) can be factored so as to yield a number of different density ratios which can then be estimated by specific discriminators; in particular, these include the appropriate marginal density ratios and then corresponding overall correction factors. As a caveat to the review I should point out that while I am familiar with GANs, they are not my main area of expertise so this should be taken into consideration - apologies if there is literature I have missed.


Experiments: The authors provide a number of illustrative experiments that demonstrate the efficacy of the approach across a number of tasks. There are many differing GAN models but due to the nature of the problem I don't have a big issue with the majority of the comparisons being against a standard GAN since the tasks are suitably designed. For the paired MNIST experiment I found it hard to assess the qualitative results visually and am always concerned about the ad-hoc nature of Inception Distances - I find it difficult to attribute weight to them quantitatively since they are usually being used to assess things where they might suffer from a common error (e.g. they are both based on NNs). Also, I'm not fully on board with the dependency metric in (5) but then the authors also point out the same concerns. The other experiments I found more convincing.

I appreciated having error bars on some of the plots to help assess significance - would it not be possible to put error bars on all plots?

I found the additional extensions presented in the appendix to be interesting ideas as well and would be interested to see how the approach works with other GAN objectives as mentioned for future work.

I am mostly very positive about this work - my main concern is really common to most GANs - all the analysis relies on the premise that the discriminators can be setup as good estimators for the density ratios. We know that this is not always the case since everything comes from samples and if the capacities of each of the discriminators are not set appropriately then I would expect problems to occur - has this been explored by the authors? It would be no detriment to the work to include failure examples where the authors purposefully make use of inappropriate architectures for some of the discriminators to check for this? For example, there will be large imbalances in the number of training samples used for the different discriminators - how does this affect stability?


Other notes:

- Whilst I understand the point about independent marginals in 2.4 I'm not sure I see the motivation as clearly since it seems that the model is much more useful when there is dependent information but maybe there's a use-case I'm not thinking of?

**Experience Assessment:**

I have read many papers in this area.

**Review Assessment: Checking Correctness Of Derivations And Theory:**

I carefully checked the derivations and theory.

**Review Assessment: Checking Correctness Of Experiments:**

I assessed the sensibility of the experiments.

**Review Assessment: Thoroughness In Paper Reading:**

I read the paper thoroughly.

---

> ### Author Response · Authors · 2019-11-14
> **Author response**
>
> We would like to thank you for your thoughtful review and are delighted about your positive assessment of the paper.
>
> "For the paired MNIST experiment I found it hard to assess the qualitative results visually and am always concerned about the ad-hoc nature of Inception Distances - I find it difficult to attribute weight to them quantitatively since they are usually being used to assess things where they might suffer from a common error (e.g. they are both based on NNs)."
>
> We agree that the evaluation metric is not necessarily optimal. Since GAN evaluation is still an unsolved problem however, we believe providing Inception distances along with visual examples is a reasonable choice given the lack of clearly superior alternatives.
>
> "I appreciated having error bars on some of the plots to help assess significance - would it not be possible to put error bars on all plots?"
>
> We included error bars wherever possible, as we agree they are quite helpful to assess significance. Unfortunately, we are not able to add them to the other plots due to the high computational requirements of training each model in each configuration (multiple days of training on a single GPU), combined with the considerable number of different configurations.
>
> "Also, I'm not fully on board with the dependency metric in (5) but then the authors also point out the same concerns. "
>
> We agree that the metric is not without flaws. However, we believe that including the metric provides useful information and thus decided to keep it in the paper.
>
> Finally, we agree that training stability is an important aspect in our setting, since we rely on the discriminators being good estimators of the respective density ratios.
> While we did not observe them in the experiments we included in the paper, we did notice that regularisation of the discriminators (here in the form of spectral normalisation) is important to ensure stability. Without such regularisation, the p-dependency discriminator can become very confident in its predictions, leading to large gradients to the generator that can prevent successful training. While we can not add further experiments easily due to the paper's space constraints, we included a short summary of this issue with a focus on how it could be resolved by extending our theoretical framework to inherently more stable GAN formulations into the conclusion section of the paper.
>
> About your note on independent marginals, it is correct that the model in this setting is more constrained than the general variant we propose. However there are some use-cases, such as independent component analysis, where an input has to be separated into components that do not exhibit dependencies between each other. This setting would be tackled in our framework by feeding the input to the generator, and viewing each output component as its own marginal, so that the q-dependency discriminator will ensure that the marginal outputs are independent.

---

### Official Review · AnonReviewer4 · 2019-11-06
**Official Blind Review #4**

**Rating:** 6

**Review:**

The paper is tackling the problem of training generative adversarial networks with incomplete data points. The problem appears to be important for semi-supervised training of image to image translation models, where we may have a lot of observations in both domains, but a little annotated correspondences between the domains.

The solution proposed by the authors involves an observation that discriminator in GANs is estimating the density ratio between real and fake distributions. This ratio can then be decomposed into a product of marginal density ratios, with two additional multipliers, corresponding to density ratios between a joint real/fake distribution and a product of its marginals. The authors then use discriminators to approximate all the ratios, which allows them to facilitate semi-supervised training.

My decision is "weak accept".

It is not clear to me to what extent does the proposed model outperform the regular CycleGAN on a large amount of paired training samples due to architectural changes (including spectral normalization).

Also, it would be nice if the comparison was carried out with a newer, possibly SotA models for unpaired image-to-image translation (MUNIT, FUNIT, BicycleGAN).

Moreover, there are some simple modifications that can be made to a standard CycleGAN/Pix2pix training pipeline that would facilitate the small number of annotations (for example, see "Learning image-to-image translation using paired and unpaired training samples").

It is hard to evaluate the comparative performance of the method without the comparisons mentioned above.

**Experience Assessment:**

I have published one or two papers in this area.

**Review Assessment: Checking Correctness Of Derivations And Theory:**

I assessed the sensibility of the derivations and theory.

**Review Assessment: Checking Correctness Of Experiments:**

I assessed the sensibility of the experiments.

**Review Assessment: Thoroughness In Paper Reading:**

I read the paper at least twice and used my best judgement in assessing the paper.

---

> ### Author Response · Authors · 2019-11-14
> **Author response**
>
> Thanks for your generally positive review and your useful feedback. We would like to respond to the questions raised in the following.
>
> "It is not clear to me to what extent does the proposed model outperform the regular CycleGAN on a large amount of paired training samples due to architectural changes (including spectral normalization)."
>
> We ran additional experiments for the CycleGAN on the image segmentation task, using the same network architectures and training setup as the GAN and FactorGAN (so that the standard GAN loss is used alongside spectral normalization). We included the results in an updated version of the paper, so please refer to the paper for more details. In short, CycleGAN is outperformed by FactorGAN in this setting, even when FactorGAN is only given 25 paired samples, and so FactorGAN is able to model the input-output dependencies more accurately. Do note however that CycleGAN treats all samples as unpaired and instead relies on its cycle consistency assumption to model the input-output dependencies.
>
> "Also, it would be nice if the comparison was carried out with a newer, possibly SotA models for unpaired image-to-image translation (MUNIT, FUNIT, BicycleGAN)."
>
> We agree that further scaling our proposed factorisation technique to more recently proposed models would be interesting. However, we believe that our main contribution is a theoretical foundation for both generation in the presence of missing data as well as general prediction tasks. It is not limited to image segmentation, and not based on a particular network architecture for the generator and discriminators, shown by the use of different networks in the paper. Therefore, we believe our experiments sufficiently support our main contribution, as they demonstrate the validity of the factorisation approach in different scenarios.
>
> "Moreover, there are some simple modifications that can be made to a standard CycleGAN/Pix2pix training pipeline that would facilitate the small number of annotations (for example, see "Learning image-to-image translation using paired and unpaired training samples")."
>
> We agree that methods such as the CycleGAN can be adapted to the same problem setting. However, many of these simple adaptations (Augmented CycleGAN, the method described in the 'Learning image-to-image translation' paper) involve adding more loss terms to the objective in an ad-hoc manner which makes it difficult to characterise optimal solutions of the overall optimisation objective. It also results in more hyper-parameters required for balancing the different loss terms. Additionally, the tasks for the discriminators can overlap – for example in the 'Learning image-to-image translation' paper, where one discriminator models the marginal generator output while another the conditional generator output. In contrast, our factorisation elegantly partitions the joint modeling task and assigns it to multiple discriminators without functional overlaps. Furthermore, as we show in the paper, we can keep the standard GAN loss where equilibrium is reached when the generator and data distribution are the same.
> To add to this, the paper you mentioned is not only restricted to deterministic generators, but also uses a cycle consistency loss that relies on the assumption that the mapping between the domains is deterministic and bijective. Since this is not the case for many problems (including the Cityscapes segmentation task), the perfect reconstruction encouraged by the cycle consistency loss is not possible. This has detrimental effects on the resulting model, as shown for the CycleGAN learning to embed extra information in its outputs to circumvent the information loss when mapping from one domain to the other that would normally make perfect reconstruction impossible. [1]
> Regardless, we included the mentioned paper in the related work section.
>
> [1] "CycleGAN, a Master of Steganography", Casey Chu, Presentation at the Machine Deception Session, NeurIPS 2017

---

### Decision · Program_Chairs · 2019-12-19

**Decision:**

Accept (Poster)

**Comment:**

All three reviewers appreciate the new method (FactorGAN) for training generative networks from incomplete observations. At the same time, the quality of the experimental results can still be improved. On balance, the paper will make a good poster.